# COMBINATORIAL ATTACKS ON BINARIZED NEURAL NETWORKS

**Elias B. Khalil**
College of Computing
Georgia Tech
lyes@gatech.edu

**Amrita Gupta**
College of Computing
Georgia Tech
agupta375@gatech.edu

**Bistra Dilkina**
Department of Computer Science
University of Southern California
dilkina@usc.edu

## ABSTRACT

Binarized Neural Networks (BNNs) have recently attracted significant interest due to their computational efficiency. Concurrently, it has been shown that neural networks may be overly sensitive to "attacks" – tiny adversarial changes in the input – which may be detrimental to their use in safety-critical domains. Designing attack algorithms that effectively fool trained models is a key step towards learning robust neural networks. The discrete, non-differentiable nature of BNNs, which distinguishes them from their full-precision counterparts, poses a challenge to gradient-based attacks. In this work, we study the problem of attacking a BNN through the lens of combinatorial and integer optimization. We propose a Mixed Integer Linear Programming (MILP) formulation of the problem. While exact and flexible, the MILP quickly becomes intractable as the network and perturbation space grow. To address this issue, we propose **IProp**, a decomposition-based algorithm that solves a sequence of much smaller MILP problems. Experimentally, we evaluate both proposed methods against the standard gradient-based attack (PGD) on MNIST and Fashion-MNIST, and show that **IProp** performs favorably compared to PGD, while scaling beyond the limits of the MILP.

## 1 INTRODUCTION

The success of neural networks in vision, text and speech tasks has led to their widespread deployment in commercial systems and devices. However, these models can often be fooled by minimal perturbations to their inputs, posing serious security and safety threats (Goodfellow et al., 2014). A great deal of current research addresses the "robustification" of neural networks using adversarially generated examples (Kurakin et al., 2016; Madry et al., 2017), a variant of standard gradient-based training that uses adversarial training examples to defend against possible attacks. Recent work has also formulated the problem of "adversarial learning" as a robust optimization problem (Madry et al., 2017; Kolter & Wong, 2017; Sinha et al., 2017), where one seeks the best model parameters with respect to the loss function as measured on the worst-case adversarial perturbation of each point in the training dataset. Attack algorithms may thus be used to augment the training dataset with adversarial examples during training, resulting in more robust models (Kurakin et al., 2016). These new advances further motivate the need to develop effective methods for generating adversarial examples for neural networks.

In this work, we focus on designing effective attacks against *Binarized* Neural Networks (BNNs) (Courbariaux et al., 2016). BNNs are neural networks with weights in $\{-1, +1\}$ and the sign function non-linearity, and are especially pertinent in low-power or hardware-constrained settings, where they have the potential to be used at an unprecedented scale if deployed to smartphones and other edge devices. This makes attacking, and consequently robustifying BNNs, a task of major importance. However, the discrete, non-differentiable structure of a BNN renders less effective the typical attack algorithms that rely on gradient information. As strong attacks are crucial to effective adversarial training, we are motivated to address this problem in the hope of generating better attacks.

The goal of adversarial attacks is to modify an input slightly, so that the neural network predicts a different class than what it would have predicted for the original input. More formally, the task of

generating an optimal adversarial example is the following:

**Given:**

- A (clean) data point $x \in \mathbb{R}^n$;
- A trained BNN model with parameters $w$, that outputs a value $f_c(x; w)$ for a class $c \in \mathcal{C}$;
- `prediction`, the class predicted for data point $x$, $\arg\max_{c \in \mathcal{C}} f_c(x; w)$;
- `target`, the class we would like to predict for a slightly perturbed version of $x$;
- $\epsilon$, the maximum amount of perturbation allowed in any of the $n$ dimensions of the input $x$.

**Find:**
A point $x' \in \mathbb{R}^n$, such that $\|x - x'\|_\infty \leq \epsilon$ and the following objective function is maximized:

$$f_{\texttt{target}}(x'; w) - f_{\texttt{prediction}}(x'; w).$$

This objective function guides *targeted* attacks (Kurakin et al., 2016), and is commonly used in the adversarial learning literature. If an adversary wants to fool a trained model into predicting that an input belongs to a given class, they will simply set the value of `target` accordingly to that given class. We note that our formulation and algorithm also work for *untargeted* attacks via a simple modification of the objective function.

Towards designing *optimal* attacks against BNNs, we propose to model the task of generating an adversarial perturbation as a Mixed Integer Linear Program (MILP). Integer programming is a flexible, powerful tool for modeling optimization problems, and state-of-the-art MILP solvers have achieved excellent results in recent years due to algorithmic and hardware improvements (Achterberg & Wunderling, 2013). Using a MILP model is conceptually and practically useful for numerous reasons. First, the MILP is a natural model of the BNN: given that a BNN uses the sign function as activation, the function the network represents is *piecewise constant*, and thus directly representable using linear inequalities and binary variables. Second, the flexibility of MILP allows for various constraints on the type of attacks (e.g. locality as in an early version of (Tjeng et al., 2017)), as well as various or even multiple objectives (e.g. minimizing perturbation while maximizing misclassification). Third, globally optimal perturbations can be computed using a MILP solver on small networks, allowing for a precise evaluation of existing attack heuristics in terms of the quality of the perturbations they produce.

The generality and optimality provided by MILP solvers does, however, come at a computational cost. While we were able to solve the MILP to optimality for small networks and perturbation budgets, the solver did not scale much beyond that. Nevertheless, experimental results on small networks revealed a gap between the performance of the gradient-based attack and the best achievable. This finding, coupled with the non-differentiable nature of the BNN, suggests an alternative: a combinatorial algorithm that is: (a) more scalable than a MILP solve, and (b) more suitable for a non-differentiable objective function.

To this end, we propose **IProp** (Integer Propagation), an attack algorithm that exploits the discrete structure of a BNN, as does the MILP, but is substantially more efficient. **IProp** tunes the perturbation vector by iterations of "target propagation": starting at a desirable activation vector in the last hidden layer $D$ (i.e. a target), **IProp** searches for an activation vector in layer $(D - 1)$ that can induce the target in layer $D$. The process is iterated until the input layer is reached, where a similar problem is solved in continuous perturbation space in order to achieve the first hidden layer's target. Central to our approach is the use of MILP formulations to perform layer-to-layer target propagation. **IProp** is fundamentally novel in two ways:

- To our knowledge, **IProp** is the first target propagation algorithm used in adversarial machine learning, in contrast to the typical use cases of training or credit assignment in neural networks (Le Cun, 1986; Bengio, 2014);
- The use of exact integer optimization methods within target propagation is also a first, and a promising direction suggested recently in (Friesen & Domingos, 2017).

We evaluate the MILP model, **IProp** and the Projected Gradient Descent method (with restarts) (PGD) (Madry et al., 2017) – a representative gradient-based attack – on BNN models pre-trained

on the MNIST (LeCun et al., 1998) and Fashion-MNIST (Xiao et al., 2017) datasets. We show that `IProp` compares favorably against PGD on a range of networks and across a set of evaluation metrics, especially with small perturbation budgets. As such, we believe that our work is a testament to the promise of integer optimization methods in adversarial learning and discrete neural networks.

This paper is organized as follows: we describe related work in Section 2, the MILP formulation in Section 3, the heuristic `IProp` in Section 4 and experimental results in Section 5. We conclude with a discussion on possible avenues for future work in Section 6.

## 2   RELATED WORK

Neural networks with the threshold (sign) activation function date back to early work on the Perceptron. However, the work of (Courbariaux et al., 2016) revived the interest in Binarized Neural Networks as a computationally cheap alternative to full-precision neural networks. This resurgence is due to an effective training algorithm for BNNs. Since then, BNNs have been used in computer vision (Rastegari et al., 2016) and high-performance neural networks (Umuroglu et al., 2017; Alemdar et al., 2017), among other domains. Notably, BNNs are amenable to extremely fast (embedded) hardware implementations (e.g. as in McDanel et al. (2017)), which may not be possible even for small full-precision networks.

Adversarial attacks against modern neural networks were first investigated in (Biggio et al., 2013; Szegedy et al., 2013). Since then, the area of "adversarial machine learning" has developed considerably. In (Szegedy et al., 2013), a L-BFGS method is used to find a perturbation of an input that leads to a misclassification. As an efficient alternative to L-BFGS, the Fast Gradient Sign Method (FGSM) was proposed in (Goodfellow et al., 2014): FGSM uses the gradient of the loss function with respect to the input to maximize the loss, a cheap operation thanks to backpropagation. Soon thereafter, Projected Gradient Descent (PGD), an iterative variant of FGSM, was shown to produce much more effective attacks (Kurakin et al., 2016; Madry et al., 2017); PGD with random restarts is the method that we will compare against in this work. Additionally, the Appendix includes a comparison of the proposed method with SPSA (Uesato et al., 2018). Other attacks have been developed for different constraints on the allowed amount of perturbation ($L_0, L_1, L_2$ norms, etc.) (Carlini & Wagner, 2017; Papernot et al., 2016; Moosavi Dezfooli et al., 2016).

Of relevance to our MILP approach are the MILP attacks against rectified linear unit (ReLU) networks of (Tjeng et al., 2017) and (Fischetti & Jo, 2018). In contrast to binarized networks, ReLU networks are differentiable almost everywhere and thus straightforwardly amenable to attacks via PGD. Galloway et al. (2017) perform an empirical evaluation of existing attack methods against BNNs and find that BNNs are more robust to gradient-based attacks than their full-precision counterparts. This finding suggests the search for more powerful attacks that exploit the discrete nature of a BNN, a key motivation for our work here. Most recently, Narodytska et al. (2017) studied the problem of *verifying* BNNs with satisfiability (SAT) solvers and MILP. In contrast to our *optimization* problem of maximizing the difference in outputs for a pair of classes, verification is a *satisfiability* problem that asks to prove that a network will not misclassify a given point, i.e. there is no objective function. As such, SAT solvers fare better than MILP solvers in BNN verification. Our `IProp` algorithm is complementary to the exact verification methods of Narodytska et al. (2017), as it can be used to quickly find a counterexample perturbation, if one exists, which would help resolve the verification question negatively.

## 3   INTEGER PROGRAMMING FORMULATION

We briefly introduce our Mixed Integer Linear Programming formulation for the BNN attack problem. As mentioned earlier, the MILP may not be scalable, but it offers insights into designing better algorithms for our problem, as is the case with our `IProp` algorithm. We operate on a trained, fully-connected, feed-forward BNN with weights $w_{l,j',j} \in \{-1, 1\}$ between each neuron $j'$ in the $(l-1)$-st layer and each neuron $j$ in the $l$-th layer. The BNN performs, at each of its $D$ hidden layers ($r$ neurons per layer), a linear transformation of the input followed by the (element-wise) application of the sign function, where $\texttt{sign}(x)$ is 1 if $x \geq 0$ and $-1$ otherwise. The output layer consists of a weighted sum of the final hidden layer's activations. In what follows, we use the notation $[D]$ to denote the set of integers from 1 to $D$, and $[C, D]$ to denote the set of integers from $C$ to $D$ inclusive.

We use the following variables to formulate the BNN attack:

- $p_j$: the perturbation in feature $j$, such that the perturbed point is $x + p$; this is a continuous variable, and the only decision variable in our formulation.

- $a_{l,j}$: the pre-activation sum for the $j$-th neuron in the $l$-th layer; for the output ($D + 1$-st) layer, $a_{D+1,\texttt{target}}$ and $a_{D+1,\texttt{prediction}}$ are equal to the output values $f_{\texttt{target}}(x'; w)$ and $f_{\texttt{prediction}}(x'; w)$ of the model for the two classes of interest.

- $h_{l,j}$: this is the activation value for the $j$-th neuron in the $l$-th layer, i.e. $h_{l,j} = 1$ if $a_{l,j} \geq 0$ and $h_{l,j} = 0$ otherwise. This is the only set of binary variables in our formulation.

In the following MILP formulation, the constraints essentially implement a forward pass in the BNN, from the perturbed input to the output layer. In particular, (2) and (3) compute the pre-activation sums, (4) and (5) are big-M constraints that assign the correct activation value $h$ given the pre-activation $a$, and (6) is the perturbation budget constraint. Note that for (4) and (5), we require the lower and upper bounds $L_{l,j}$ and $U_{l,j}$ on $a_{l,j}$; those bounds are easily calculated given $x$ and $\epsilon$. We implicitly assume that the input is in $[0, 1]^n$, and constrain the perturbed point to be within this range; this is typical for images for example, where pixels in $[0, 255]$ are scaled to $[0, 1]$.

$$\max \quad a_{D+1,\texttt{target}} - a_{D+1,\texttt{prediction}} \tag{1}$$

$$\text{subject to} \quad a_{1,j} = \sum_{j'=1}^{n} w_{1,j',j} \cdot (x_{j'} + p_{j'}) \qquad \forall j \in [r] \tag{2}$$

$$a_{l,j} = \sum_{j'=1}^{r} w_{l,j',j} \cdot h_{l-1,j'} \qquad \forall l \in [2, D+1], \forall j \in [r] \tag{3}$$

$$a_{l,j} \leq U_{l,j} \cdot \frac{(h_{l,j} + 1)}{2} \qquad \forall l \in [D], \forall j \in [r] \tag{4}$$

$$a_{l,j} \geq L_{l,j} \cdot \frac{(1 - h_{l,j})}{2} \qquad \forall l \in [D], \forall j \in [r] \tag{5}$$

$$p_j \in [-\epsilon, \epsilon] \qquad \forall j \in [n] \tag{6}$$

$$h_{l,j} \in \{-1, 1\} \qquad \forall l \in [D], \forall j \in [r] \tag{7}$$

$$a_{l,j} \in [L_{l,j}, U_{l,j}] \qquad \forall l \in [D+1], \forall j \in [r] \tag{8}$$

In implementing this formulation, we accommodate "batch normalization" (Ioffe & Szegedy, 2015), which has been shown to be crucial to the effective training of BNNs (Courbariaux et al., 2016). We simply use the parameters learned for batch normalization, as well as the mean and variance over the training data, to compute this linear transformation.

## 4   IPROP: INTEGER TARGET PROPAGATION

As we will see in Section 5, solving the MILP attack model becomes difficult very quickly. On the other hand, gradient-based attacks such as PGD are efficient (one forward and backward pass per iteration), but not suitable for BNNs: a trained BNN represents a piecewise constant function with an undefined or zero derivative zero at any point in the input space. This same issue arises when training a BNN. There, (Courbariaux et al., 2016) propose to replace the sign function activation by a differentiable surrogate function $g$, where $g(x) = x$ if $x \in [-1, 1]$ and $\texttt{sign}(x)$ otherwise. This surrogate function has derivative 1 with respect to $x$ between $-1$ and 1, and 0 almost everywhere else. As such, during backpropagation, PGD uses the approximate BNN with $g$ as activation, computing its gradient w.r.t. the input vector, and taking an ascent step to maximize the objective (1).

However, as we show in Figure 1, the gradient used by PGD may not be indicative of the correct ascent direction. Figure 1 illustrates the outputs of a BNN (left) and an approximate BNN (right) with 3 hidden layers and 30 neurons per layer, as a single input value is varied in a small range. Clearly, the approximate BNN can behave arbitrarily differently, and gradient information with respect to the input dimension being varied is not very useful for our task.

Motivated by this observation, as well as the limitations of MILP solving, we propose **IProp**, a BNN attack algorithm that operates directly on the original BNN, rather than an approximation of it. To gain intuition as to how **IProp** works, it is useful to reason about the form of an optimal solution to our problem. In particular, the objective function (1) can be expanded as follows:

$$a_{D+1,\text{target}} - a_{D+1,\text{prediction}} = \sum_{j=1}^{r} \left( w_{D+1,j,\text{target}} - w_{D+1,j,\text{prediction}} \right) \cdot h_{D,j}.$$

Here, the summation is over the $r$ neurons in layer $D$, and $h_{D,j} \in \{-1, 1\}$ is the activation of neuron $j$ in the last hidden layer $D$. Clearly, whenever the weights of a neuron $j$ into the two output neurons of interest are equal, i.e. $w_{D+1,j,\text{target}} = w_{D+1,j,\text{prediction}}$, the activation value of that neuron does not contribute to the objective function. Otherwise, if $w_{D+1,j,\text{target}} \neq w_{D+1,j,\text{prediction}}$, then an *ideal setting* of the activation $h_{D,j}$ would be $+1$ or $-1$, since this increases the objective function. Applying the same logic to all neurons in hidden layer $D$, we obtain an *ideal target* activation vector $\overline{T} \in \{-1, 1\}^r$ which maximizes the objective. However, $\overline{T}$ may not be achievable by any perturbation to input $x$, especially if the perturbation budget $\epsilon$ is sufficiently small. As such, **IProp** aims at achieving as many of the ideal target activation values as possible, given $\epsilon$.

**IProp** is summarized in pseudocode below. However, we invite the reader to return to the pseudocode following Section 4.3, as a lot of the notation is only introduced there.

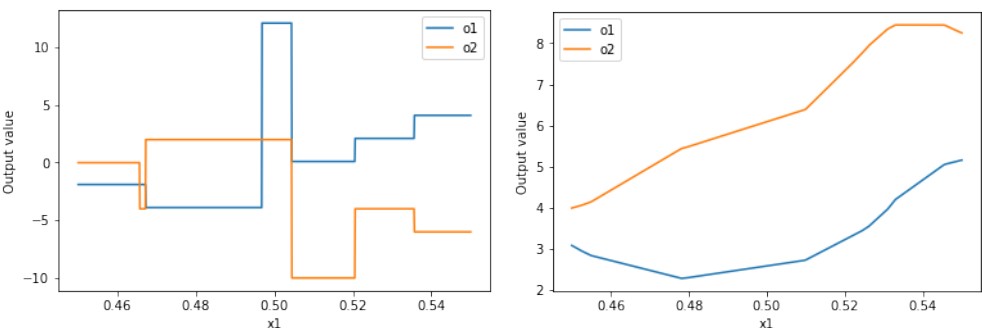

Figure 1: Final layer activations for inputs to a small BNN with two output classes (o1 and o2) as a single input dimension (x1) is varied. The relative activations of the two classes differ significantly between the true BNN (left) and an approximation of the BNN (right) used to enable gradient computations for PGD.

---

**IProp** $(x, \epsilon, \text{BNN weight matrices } \{W_l\}_{l=1}^{D}, \texttt{prediction}, \texttt{target}, \text{step size } S)$

1: Incumbent perturbation: $p^* \leftarrow \vec{0}$ (no perturbation)
2: Compute $\overline{T} \in \{-1, 1\}^r$, the ideal target activation vector in layer $D$
3: Run $x$ through BNN; Set $h_l^*$ to resulting activations in layer $l$ for all layers, and
4: $I^* = \{k \in [r] | h_D^*(k) = \overline{T}(k)\}$
5: $t = 1$
6: **while** time limit not reached and not at local optimum **do**
7:   Sample a set of $S$ neurons $G_D^t \subseteq \{k \in [r] | h_D^*(k) \neq \overline{T}(k)\}$ for layer $D$
8:   $T_D^t := I^* \cup G_D^t$
9:   **for** layer $l = (D-1)$ to 1 **do**
10:    $T_l^t = \arg\max_{h_l \in \{-1,1\}^r} \sum_{j \in T_{l+1}^t} \mathbb{I}\{h_{l+1,j} = T_{l+1}^t(j)\}$ s.t. $h_{l+1} = \text{sign}(W_{l+1}h_l)$
11:   $p^t = \arg\max_{p \in [-\epsilon,\epsilon]^n} \sum_{j=1}^{r} \mathbb{I}\{h_{1,j} = T_1^t(j)\}$ s.t. $h_1 = \text{sign}(W_1(x+p)), 0 \leq x + p \leq 1$
12:   **if** a forward pass with solution $x + p^t$ improves objective (1): **then**
13:     Update incumbent: $p^* \leftarrow p^t$; Update $h_l^*, I^*$
14:   $t = t + 1$
    **return** $p^*$

---

### 4.1 LAYER-TO-LAYER TARGET SATISFACTION

Given the ideal target $\overline{T}$, one can ask the following question: how should we set the activation vector $T_{D-1}$, which consists of the activation values $h_{D-1,j}$ in layer $(D-1)$, such that as much of $\overline{T}$ is achieved after applying the linear transformation and the sign activation? This is a *constraint satisfaction problem* with linear inequalities. More generally, if we would like a given neuron's activation $h_{l,j}$ to be equal to 1, then the corresponding $a_{l,j}$, defined in (3), must be greater than or equal to 0, and vice versa for $h_{l,j}$ to be $-1$. We cast this binary linear optimization problem as follows:

$$T_l := \underset{h_l \in \{-1,1\}^r}{\arg\max} \sum_{j=1}^r \mathbb{I}\{h_{l+1,j} = T_{l+1}(j)\} \text{ s.t. } h_{l+1} = \text{sign}(W_{l+1}h_l). \tag{9}$$

The variables to optimize over in (9) are $h_l \in \{-1,1\}^r$, whereas $T_{l+1} \in \{-1,1\}^r$ is fixed, as it is provided by the layer $(l+1)$; we describe this in detail in Section 4.2. For instance, when $l = D-1$ and $T_{l+1} = \overline{T}$, the optimization problem in (9) models the satisfaction problem described in the last paragraph.

### 4.2 TARGET PROPAGATION

Consider solving a sequence of optimization problems based on (9), starting with $l = D-1$ and ending with $l = 1$, where each solution $T_l$ to the problem at layer $l$ provides the target for the subsequent problem at layer $(l-1)$. Then, after obtaining $T_1$ as a solution to the last optimization problem in the aforementioned sequence, one can search for a perturbation of $x$ that produces $T_1$, by solving the following mixed binary program:

$$p = \underset{p' \in [-\epsilon, \epsilon]^n}{\arg\max} \sum_{j=1}^r \mathbb{I}\{h_{1,j} = T_1(j)\} \text{ s.t. } h_1 = \text{sign}(W_1(x + p')), 0 \le x + p' \le 1. \tag{10}$$

After computing the perturbation $p$, the point $(x + p)$ is run through the network, and the corresponding objective value (1) is computed. The procedure we just described is, at a high-level, a single iteration of our proposed **IProp** algorithm. We will describe the full iterative algorithm in Section 4.3.

In theory, both optimization problems (9) and (10) are NP-Hard, by reduction from the MAX-SAT problem, and thus as hard as our MILP problem of Section 3. However, in practice, problems (9) and (10) are much easier to solve than the MILP of Section 3, since they are smaller (involving a single hidden layer). We find that for networks with 2-5 hidden layers and 100-500 neurons, these layer-to-layer problems are solved optimally in a few seconds by a MILP solver. It is for this reason that we view **IProp** as a *decomposition* algorithm, in that it decomposes the full-network MILP of Section 3 into smaller subproblems (9) and (10).

However, the current description of **IProp** raises two critical questions:

1. When solving problem (9) at the last hidden layer, $l = D$, aiming to set $h_{D,j} = T_D(j)$ for *all* neurons may be overly ambitious: if $\epsilon$ is very small, then the target propagation is bound to fail when problem (10) is solved.

2. In solving the sequence of problems (9), a layer $l$'s problem may have multiple optimal solutions that achieve the same number of targets in layer $(l+1)$. What solutions should we then prefer?

Both of the questions we raised effectively relate to the perturbation budget $\epsilon$: as **IProp** decomposes the attack into layer-to-layer problems (9) and (10), it is easy to lose track of the global constraint $\epsilon$, which makes many targets $T_l$ impossible to achieve. The solutions that we describe next make **IProp** $\epsilon$-aware, and thus practically effective.

### 4.3 TAKING SMALL STEPS

To address the first question, we take inspiration from gradient optimization methods, which take small steps as determined by a step size (or learning rate), so as to not overshoot good solutions.

When solving problem (9) at the last hidden layer, we restrict the summation in the objective function to a subset of all neurons; this has the effect of only rewarding target satisfaction up to a limit, so as to not produce overly optimistic solutions that will not withstand the bound $\epsilon$. Specifically, let $p^*$ denote the current incumbent perturbation, initialized to the zero-perturbation vector. Let $h_l^*$ denote the binary activation vector of layer $l$ when the incumbent solution $(x + p^*)$ is run through the BNN. At each iteration $t$ of **IProp**, we solve the sequence of problems (9) and then (10). To do so, we must specify a set of targets for the first problem (9) that is solved at $D$. This set of targets $T_D^t$ is the union of two sets: the set $I^* = \{k \in [r]|h_D^*(k) = \overline{T}(k)\}$ of already-ideal neurons; and a small set $G^t \subseteq \{k \in [r]|h_D^*(k) \neq \overline{T}(k)\}$ of neurons who are **not** at their ideal activations under the incumbent. If $S$ denotes the step size, then $|G^t| = S$ for all $t$. In our implementation, $G^t$ is sampled uniformly and without replacement from all possible $S$-subsets of non-ideal neurons.

Importantly, after the target $T_D^t$ is specified, target propagation is performed and a potential perturbation $p^t$ is obtained and then run through the BNN. If the objective function (1) improves, the incumbent $p^*$ is updated to $p^t$, and so is the set $I^*$. In the next iteration, a new target $T_D^{t+1}$ is attempted, and **IProp** terminates when it hits a local optimum or runs out of time.

**IProp** is summarized in pseudocode above, with all intermediate optimization problems included, and using common notation.

## 4.4 MAXIMAL TARGETING AT MINIMUM COST

Having presented the full **IProp** algorithm, we now address the second question posed at the end of Section 4.2: how do we prioritize equally good solutions to problems (9)? Intuitively, if two solutions $T_l'$ and $T_l''$ have the same objective value, i.e. satisfy the same number of neurons in layer $(l + 1)$, then we would rather use the one which is "closest" to $h_l^*$, the binary activation vector of layer $l$ under incumbent solution $(x + p^*)$. Such a solution of minimum cost, in the sense of minimum deviation from the forward pass activations of the incumbent, is likely to be easier to achieve when layer $(l-1)$'s problem (9) is solved. As a cost metric, we use the $L_0$ distance between $h_l^*$ and the variables $h_l$. Note that this cost metric is used as a tie-breaker, and is incorporated into the objective of (9) directly with a small multiplier, guaranteeing that the original objective of (9) is the first priority. We omit this term from the **IProp** pseudocode above for lack of space.

## 5 EXPERIMENTS

To train the binarized neural networks for which we generate attacks, we use BNN code [1] by Courbariaux et al. (2016), and run training experiments on a machine equipped with a GeForce GTX 1080 Ti GPU. We train networks with the following depth x width values: 2x100, 2x200, 2x300, 2x400, 2x500, 3x100, 4x100, 5x100. While these networks are not large by current deep learning standards, they are larger than most networks used in recent papers (Fischetti & Jo, 2018; Narodytska et al., 2017) that leverage integer programming or SAT solving for adversarial attacks or verification. All BNNs are trained to minimize the cross-entropy loss with "batch normalization" (Ioffe & Szegedy, 2015) for 100 epochs on the full 60,000 MNIST and Fashion-MNIST training images, achieving between 90–95% test accuracy on MNIST, and 80–90% on Fashion-MNIST.

For attack generation, we use the Gurobi Python API to implement and solve our MILP problems, and an implementation of iterated PGD in PyTorch. All methods are run with a time cutoff of 3 minutes on 1,000 test points from the MNIST dataset and 100 test points from the Fashion-MNIST dataset. The MILP problems (9), (10) solved within **IProp** are given a 10 second cutoff. All attacks are run on a cluster of 5 compute nodes, each with 64 cores and 256GB of memory. In the experiments that follow, we specify the class with the second-highest activation (according to the trained model) on the original input as the target class.

## 5.1 GENERATING ADVERSARIAL EXAMPLES

Figure 2 shows the fraction of MNIST and Fashion-MNIST test points that were flipped by a given attack, for a given network (depth, width) and perturbation budget $\epsilon$; a flip occurs when the objec-

---

[1]`https://github.com/itayhubara/BinaryNet.pytorch/`

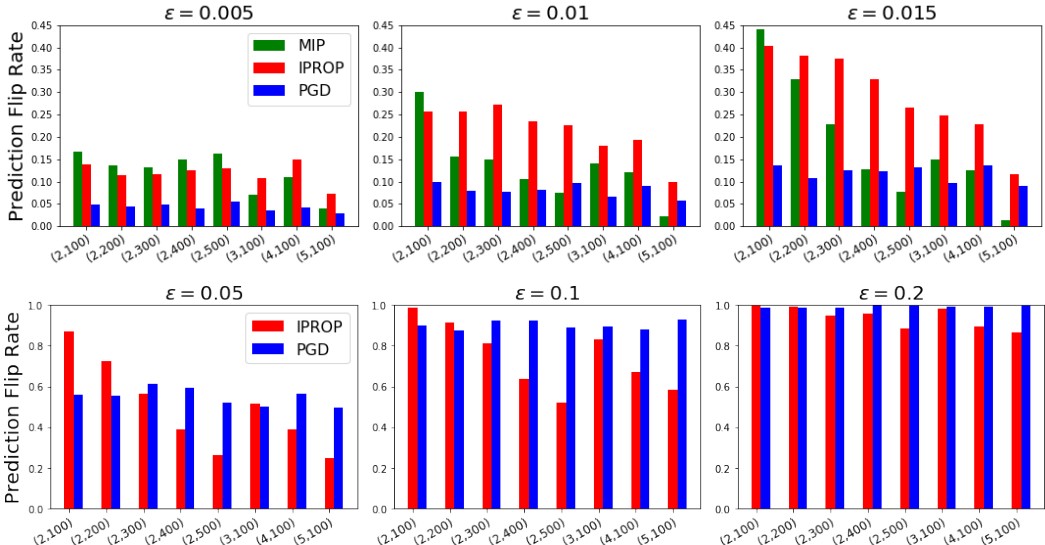

Figure 2: Proportion of samples for which the final prediction was flipped to the target class (y-axis) by MIP vs. PGD vs. **IProp** attacks with varying network architectures (x-axis) and varying $\epsilon$ (left-right), on the MNIST dataset.

tive (1) is strictly positive. A higher value is better here. We compare attacks generated using MILP, our method, and PGD on samples from MNIST. For small perturbation budgets $\epsilon$ and networks, the MILP approach finds optimal attacks within the time cutoff, but as $\epsilon$ and network size grow, solving the MILP becomes increasingly computationally intensive and only the best-found solution at time-out is returned. Specifically, for the 2x100 network with $\epsilon = 0.01$, the average runtime of the solver is 27 seconds (all test instances solved to optimality), whereas the same quantity is 777 seconds for the 2x200 network for the same value of $\epsilon$. Similar behavior can be observed as $\epsilon$ grows, with most runs timing out at the MILP time limit of 1800 seconds. We believe that this is largely due to the weakness of the linear programming relaxation, as observed by Fischetti & Jo (2018), and perhaps the mismatch between the kind of heuristics Gurobi implements versus what would be useful for neural network problems such as ours.

Our method, **IProp** (in red bars), achieves a success rate close to the optimal MILP performance on small networks and $\epsilon$, and scales better than the MILP approach. **IProp** outperforms PGD for nearly all network architectures for the three smaller $\epsilon$ values. The better performance of **IProp** compared to PGD is of particular interest for small perturbations, as these are more challenging to detect as attacks. Note that the inputs are in $[0, 1]$, and so $\epsilon = 0.005$ corresponds to a 0.5% change in pixel intensity. For larger values of $\epsilon$, fooling the BNN is relatively easy, as manifested by the high bars. PGD can outperform **IProp** in this easy regime since **IProp** is more computationally expensive. Figure 4, shows box plots of the (normalized) objective value (1) across the different settings. Consistently with Figure 2, **IProp** achieves higher values on average than PGD, indicating that the **IProp** attacks are more effective at modifying the output-layer activations of the networks.

One might wonder about the behavior of the **IProp** and PGD attack methods over time, as PGD is widely regarded as a fast, reasonably-effective attack method. Figure 5 shows the relative solution quality over time for each method, averaged over MNIST samples. It is evident that iterated PGD ceases to improve greatly after the first 30 seconds or so. However, more effective attacks are clearly possible, and the **IProp** algorithm constructs progressively stronger attacks that typically surpass the best found PGD attacks after a few more seconds.

## 5.2 ANALYSIS OF **IPROP**

Additionally, we investigate the effect of step size $S$ in Line 7 of **IProp** (Figure 6). Intuitively, using a small step size $S$ may ensure that the target activations used in each successive iteration are not too difficult to achieve from the current activation in layer $D$, but this may also lead to multiple

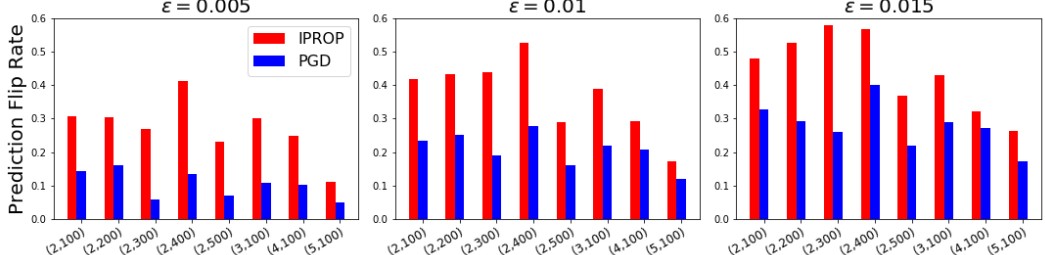

Figure 3: Proportion of samples for which the final prediction was flipped to the target class (y-axis) by PGD vs. **IProp** attacks with varying network architectures (x-axis) and varying $\epsilon$ (left-right), on the Fashion-MNIST dataset.

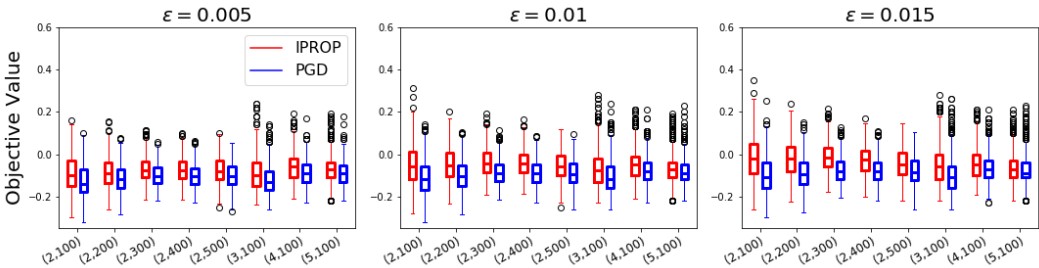

Figure 4: Summary statistics for the normalized objective value of attacks obtained by **IProp** versus PGD (y-axis) with varying $\epsilon$ in networks with different architectures, on MNIST.

iterations and slow improvement over time. Another consideration is that for small perturbation budgets $\epsilon$, large changes in the layer $D$ target activation may propagate back to the first hidden layer, only to fail at the input layer. Meanwhile, wider network architectures may permit the use of larger step sizes. To that end, we devise an *adaptive* step size strategy ("Adaptive", red in all figures): initialized at 5% of the width of the network, the step size $S$ is halved every 5 iterations, if no better incumbent is found. While the hyperparameters of this strategy (initial value, decay rate and number of iterations before decaying) may be optimized, the set of values we used performed reasonably well, as can be seen in Figure 6. Indeed, for many of the settings shown, "Adaptive" performs best or close to the best fixed "Constant" step size. Note that previous figures showing **IProp** in red correspond to this very adaptive step size strategy.

One minor modification that highlights the flexibility of the **IProp** attack method is our ability to warmstart the algorithm with an initial perturbation. For example, we used perturbations obtained by running PGD with a time cutoff of 5 seconds as an alternative to using no perturbation in Line 1 of **IProp**. Figure 7 shows that warm starting **IProp** in this manner has the potential to significantly improve the success rate of the resulting attacks, highlighting the value of finding good initial solutions our method, which is essentially a combinatorial local search approach.

## 6 CONCLUSION & DISCUSSION

We developed combinatorial search methods for generating adversarial examples that fool trained Binarized Neural Networks, based on a Mixed Integer Linear Programming (MILP) model and a target propagation-driven iterative algorithm **IProp**. To our knowledge, this is the first such integer optimization-based attack for BNNs, a type of neural networks that is *inherently discrete*. Our MILP model results show that standard (PGD) attack methods often are suboptimal in generating good adversarial examples when the perturbation budget is limited. The ultimate goal is to "attack to protect", i.e. to generate perturbations that can be used *during adversarial training*, resulting in BNNs that are robust to a class of perturbation. Unfortunately, our MILP model cannot be solved quickly enough to be incorporated into adversarial training. On the other hand, through extensive experiments we have shown that our iterative algorithm **IProp** is able to scale-up this solving

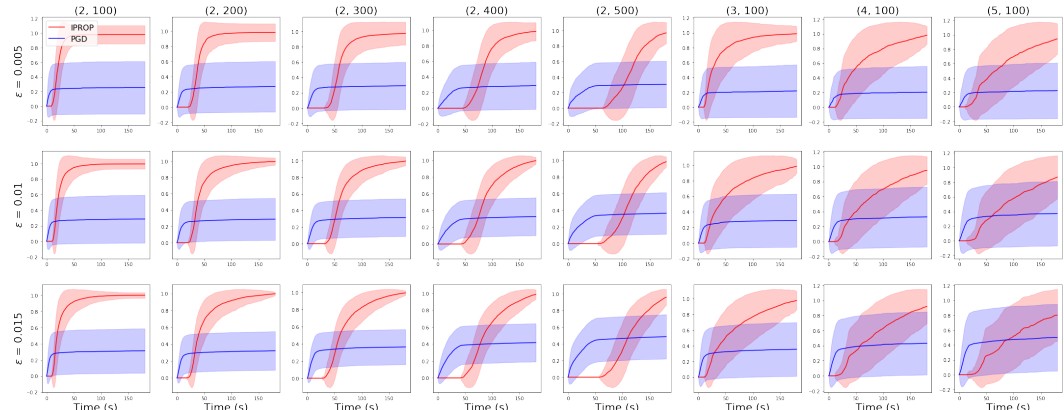

Figure 5: Average normalized solution objective value (y-axis) versus runtime (x-axis) for **IProp** versus PGD on MNIST samples.

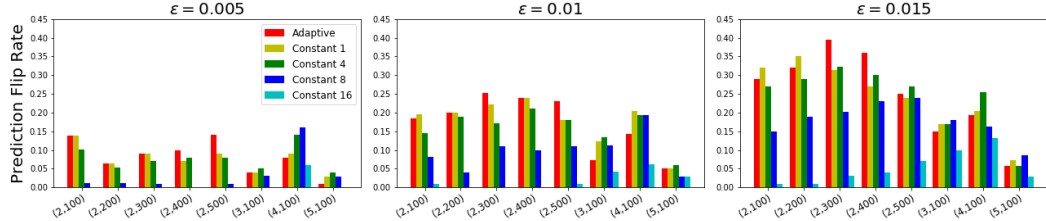

Figure 6: Proportion of MNIST samples on which the final prediction was flipped to the target class by **IProp** with adaptive or constant step sizes. The adaptive step size performs relatively well across networks of varying size and different values of $\epsilon$.

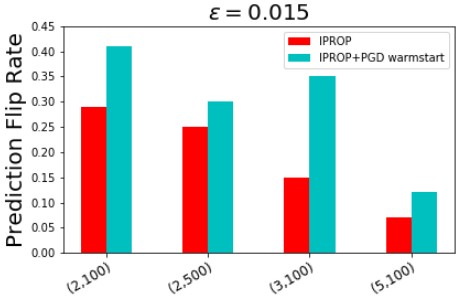

Figure 7: Proportion of MNIST samples on which the final prediction was flipped to the target class by **IProp** starting with zero perturbation or with an initial perturbation found by running PGD for a short amount of time.

process while maintaining good performance compared to the PGD attack. With these contributions, we believe we have laid the foundations for improved attacks and potentially robust training of BNNs. This work is a good example of successful cross fertilization of ideas and methods from discrete optimization and machine learning, a growing synergistic area of research, both in terms of using discrete optimization for ML as was done here (Friesen & Domingos, 2017; Bertsimas et al., 2017; Bertsimas & Van Parys, 2017; Anderson et al., 2018), as well as using ML in discrete optimization tasks (He et al., 2014; Sabharwal et al., 2012; Khalil et al., 2016; Kruber et al., 2016; Dai et al., 2017). We believe that target propagation ideas such as in **IProp** can be potentially extended for the problem of *training* BNNs, a challenging task to this day. The same can be said about hard-threshold networks, as hinted to by Friesen & Domingos (2017).

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

APPENDIX

## 6.1 COMPARISON TO SPSA

We implemented the method of "simultaneous perturbation stochastic approximation" (SPSA) (Spall et al., 1992), which was recently used in (Uesato et al., 2018) as an example of a gradient-free attack. Our implementation of SPSA follows (Uesato et al., 2018) and uses the Adam optimization method with learning rate 0.01, a stochastic sample of perturbations (referred to as "batch size" in (Uesato et al., 2018)) of size 100, and an iteration limit of 100. As with PGD, SPSA is run with random restarts every 100 iterations until the time limit of 180 seconds is reached.

Figure 8 shows the flip prediction rates for **IProp** (same as in Figure 2 in the main text) and SPSA. Generally, SPSA performs worse than **IProp** and PGD.

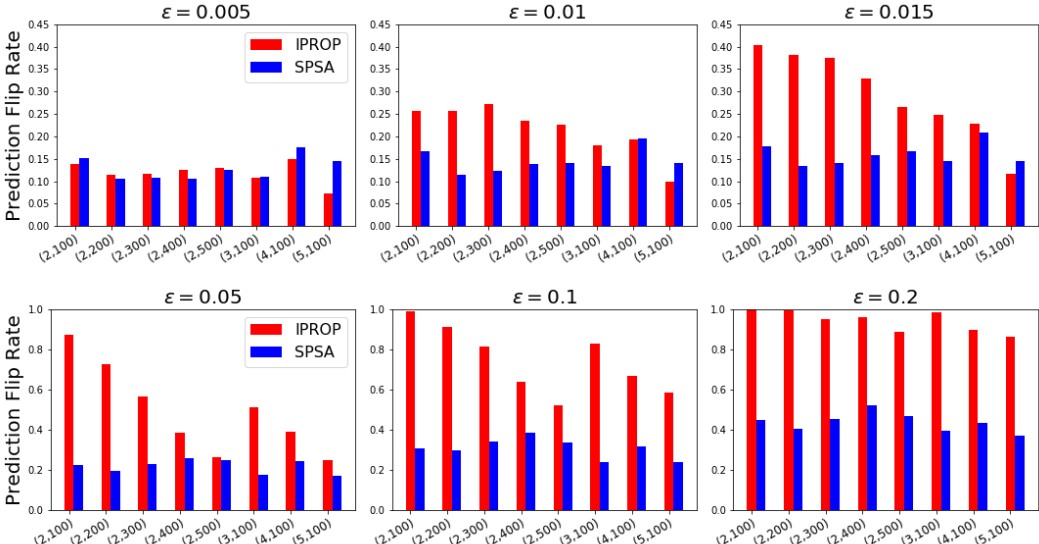

Figure 8: Proportion of samples for which the final prediction was flipped to the target class (y-axis) by SPSA vs. **IProp** attacks with varying network architectures (x-axis) and varying $\epsilon$ (left-right), on the MNIST dataset.

