# OpenReview forum: "Combinatorial Attacks on Binarized Neural Networks"
_ICLR.cc/2019/Conference_

### Official Review · AnonReviewer3 · 2018-11-02
**Reviewer comments**

**Rating:** 7
**Confidence:** 4

**Review:**

This paper presents an algorithm to find adversarial attacks to binary neural networks.  Binary neural networks uses sign functions as nonlinearities, making the network essentially discrete.  Previous attempts at finding adversarial attacks for binary neural networks either rely on relaxation which cannot find very good adversarial examples, or calling a mixed integer linear programming (MILP) solver which doesn’t scale.  This paper proposes to decompose the problem and iteratively find desired representations layer by layer from the top to the input.  This so called Integer Propagation (IProp) algorithm is more efficient than solving the full MILP as it solves much smaller MILP problems, one for each layer, thus each step can be solved relatively quickly.  The authors then proposed a few more improvements to the IProp algorithm, including ways to do local adjustments to the solutions, and warming starting from an existing solution.  Experiments on binary neural nets trained for MNIST and Fashion MNIST show the superiority of the proposed method over MILP and relaxation based algorithms.

Overall I found the paper to be very clear and the proposed method is sound.  I think combining ideas from discrete / combinatorial optimization with deep learning is an important research direction and can shed light on training and verifying models with discrete components, like the hard nonlinearities in the binary neural nets studied in this paper.

In terms of the particular proposed approach, it is hard for me to imagine the blind IProp that does not take the input into account until the last layer is ever going to work.  The small step size modifications make a lot more sense.  Regarding the selection of the set S, in the paper the authors simply sampled elements to be in S uniformly, but it seems possible to make use of the information from the forward pass, and choose the hidden units that are the closed to reaching the desired activations.  Would that be any better?

A few minor comments:
- when reporting warm start results, it would be good to also show the performance of the FGSM solution used for warm starting, in addition to the other two results shown in Figure 6 to have a more complete comparison
- the hidden units h_{l,j} were formulated to be in {0, 1} in equation (7), but everywhere else in the paper they are assumed to be in {-1, +1}, which is not consistent and slightly confusing.

Overall I think this is a solid paper and support accepting it for publication.

---

> ### Author Response · Authors · 2018-11-14
> **Response to Reviewer 3**
>
> Thank you for the positive comments and suggestions!
> Regarding the set S: indeed, your suggestion is valid and we have tried it early on. We sampled neurons closer to the threshold (zero) with higher probability than others. We did not observe much improvement over uniform sampling at the time, and thus decided to stick with simple uniform sampling.
>
> Regarding warmstart results: that’s a great point; we will do so in the final version of the paper.
>
> Regarding notation: thanks for catching that; we will make the notation consistent throughout.

---

### Official Review · AnonReviewer2 · 2018-11-04
**an interesting paper**

**Rating:** 6
**Confidence:** 4

**Review:**

This paper proposed a new attack algorithm based on MILP on binary neural networks. In addition to the full MILP formulation, the authors proposed an integer target propagation algorithm (IProp) to find adversarial examples by solving a smaller (instead of the full) MILP.

The topic is important but the clarity should be improved. It is less clear when describing the Iprop algorithm.

Questions:
1. Can IProp work for other architectures? It looks like the propagation steps work on only fully connected layers (or conv layers) with activation functions. Does it work for pooling layers?
2. The results in Figure 2 look weird and might be wrong:
since MIP is the exact solution (green bar), how is it possible that the prediction flip rate of IProp larger than MIP? See top row figures where some red bars are larger than green bars.
3. Also, is the FGSM method comparing in Figure 2 operating on the approximate BNN as described in the related work? How does the performance of PGD (Madry etal) compared to IProp?
4. How are the big M parameters in equation 4 and 5 computed? Is the formulation eq (1) to (8) the same as that in Tjeng 2018? Since BNN is a special case of general neural networks. Please elaborate.
5. In Sec 2 related work, why "there's no objective function" for verification method?

---

> ### Author Response · Authors · 2018-11-14
> **Response to Reviewer 2**
>
> Thank you for taking the time to review our paper. Our answers to your questions are numbered in the same order as your review:
>
> 1. Yes, IProp does work for pooling layers, as the layer-to-layer satisfaction problem (section 4.1) can be modified to compute a pooling transformation by adding constraints appropriately. For instance, max/mean pooling are easily implemented with linear inequalities and/or binary variables.
>
> 2. we discuss this point at length in section 5.2, page 7. The MIP solver fails to scale to the wider/deeper networks, and thus times out at the 3-minute cutoff. The final solution returned by MIP may thus be suboptimal, which results in green bars being smaller than red bars.
>
> 3. (same reply as to reviewer 1) Thanks for raising this point - we already use PGD and will clarify this in writing. In our paper, FGSM refers to “iterated FGSM” or “multi-step FGSM” or PGD (these are all referring to the same method, e.g. see page 4 of https://arxiv.org/abs/1706.06083). We make that clear in section 2: “Soon thereafter, an iterative variant of FGSM was shown to produce much more effective attacks (Kurakin et al., 2016); it is this version of FGSM that we will compare against in this work.”. In fact, we run iterated FGSM/PGD for 3 minutes (same as MIP and IProp) with random restarts every 100 iterations. This provides FGSM with the same computational budget as IProp. We will update the paper to clarify this point in the experiments section.
>
> 4. the big-M values are computed by simply bounding the a_{1,j} variables at the first hidden layer, since the input image is in an epsilon-box. Then, those bounds are passed on to the h_{1,j} variables, i.e. if the lower and upper bounds on a given a_{1,j} are negative, then h_{1,j} must be -1. Those bounds on h_{1,j} are then propagated to the a_{2,j} variables, and so on and so forth. This procedure is simple and runs in time linear in the size of the network. We are happy to describe it in the paper, if the reviewer thinks that would be useful.
> Our formulation differs from that of Tjeng in that our constraints (4), (5) and (7) encode the discrete sign activation function and the binary weights.
>
> 5. in Narodytska et al. (2018), the goal is to prove that an input to a network cannot be fooled with epsilon perturbations, or provide a counter-example to that. As such, they do not care about maximizing the difference between the incorrect class and the true class as we do. In other words, the verification problem in Narodytska et al. (2018) is a feasibility problem rather than an optimization problem, and so it does not have an explicit objective function.

---

### Official Review · AnonReviewer1 · 2018-11-04
**Interesting and novel idea, needs more experimental validation**

**Rating:** 5
**Confidence:** 4

**Review:**

The authors study the problem of generating strong adversarial attacks on binarized neural networks (networks whose weights are binary valued and have a sign function nonlinearity).  Since these networks are not continuous (due to the sign function nonlinearity), it is possible that standard gradient-based attack algorithms are not effective at producing adversarial examples. While this problem can be encoded as a mixed integer linear program, off-the-shelf MILP solvers are not scalable to larger/deeper networks. Thus, the authors propose a new target propagation style algorithm that attempts to infer desired activations at each layer (from the perspective of maximizing the adversary's objective) starting at the final layer and moving towards the input. The propagation at each layer requires solving another MILP (albeit a much smaller one). Further, in order to prevent the target propagation from discovering assignments at upper layers that are unachievable given the constraints at lower layers, the authors propose two heuristics (making small moves and penalizing deviations from the previous target values) to obtain an effective attack algorithm. The authors validate their approach experimentally on MNIST/Fashion MNIST image classifiers.

Quality: The paper is reasonably well written and the key ideas are communicated well. However, the experimental section needs to be improved significantly.

Clarity: The paper is easy to understand and organized well.

Originality: The application of target propagation in the context of adversarial examples is certainly novel and so are the specific enhancements proposed in the context of adversarial example generation. The

Significance: The study of adversarial examples for binarized networks is novel and important and effective attack generation algorithms are a significant first step towards training robust models of this type - this could enable deployment of robust and compact binarized classifiers in on-device settings (where model size is important).

Cons
My main concerns with this paper are regarding the experimental evaluation - I do not feel these are sufficient to justify the strength of the attack method proposed. Here are my broad concerns:
1. Even though the datasets used are small (MNIST/Fashion MNIST), the experimental validation of adversarial attacks is only performed on 100 test examples. This is not sufficiently representative (given experimental evidence with adversarial attacks on non-binarized models) and this needs to be addressed for the results to be considered conclusive.

2. The attack method is only compared to FSGM, which is known to be a rather poor attack even on non-binarized networks. The authors should compare to stronger gradient based attacks (like PGD) and gradient free attacks which have been used to break adversarial defenses that are nondifferentiable in prior work - https://arxiv.org/abs/1802.00420 and https://arxiv.org/abs/1802.05666). Further, the MILP approach used can be strengthened by doing better bound propagation (like in https://arxiv.org/pdf/1711.00455.pdf)

3. The attack radii used are very small compared to what has been used in non-binarized networks, where networks have been trained to even be verifiably robust to adversarial pertrubations of much larger radii (see for example https://arxiv.org/pdf/1805.12514.pdf). Given the existence of this work, it is important to evaluate the algorithms proposed on larger radii (since it is possible to construct non-binarized networks that are indeed robust to perburbations of eps=.1-.3 on MNIST).

4. Motivation for binarization: I assume that motivation for binarized models arising from faster training/inference times and smaller model sizes. However, to justify this, the authors need to compare their BNNs to comparable non-binarized neural networks (for example,ones that are similar  in terms of number of bits used to represent the model) on training time, inference time and adversarial robustness. Otherwise, it seems hard to see why binarized networks are valuable from a robustness.

---

> ### Author Response · Authors · 2018-11-14
> **Response to Reviewer 1**
>
> Thanks for the detailed comments - we believe most of your concerns are clarified below. In particular, our FGSM is the same as the PGD you refer to, as we explain below.
>
> 1. We are currently running the same experiments reported in the paper on a much larger set of test images, and will report the updated results as soon as they become available.
>
> 2.
> - Regarding PGD: Thanks for raising this point - we already use PGD and will clarify this in writing. In our paper, FGSM refers to “iterated FGSM” or “multi-step FGSM” or PGD (these are all referring to the same method, e.g. see page 4 of https://arxiv.org/abs/1706.06083). We make that clear in section 2: “Soon thereafter, an iterative variant of FGSM was shown to produce much more effective attacks (Kurakin et al., 2016); it is this version of FGSM that we will compare against in this work.”. In fact, we run iterated FGSM/PGD for 3 minutes (same as MIP and IProp) with random restarts every 100 iterations. This provides FGSM with the same computational budget as IProp. We will update the paper to clarify this point in the experiments section.
>
> - Regarding gradient-free attacks: Thanks for bringing those papers to our attention. The first paper (https://arxiv.org/abs/1802.00420) proposes a method that uses the straight-through estimator to approximate gradients of a non-differentiable network; this is indeed the same trick used for FGSM/PGD on BNNs, and so our comparison with PGD already covers the method BPDA proposed in the paper. Regarding the second paper (https://arxiv.org/pdf/1802.05666.pdf), we are now implementing it and will report on results as soon as they become available.
>
> - Regarding bound propagation: indeed, we already do perform bound propagation since the input images are bounded in a small epsilon-box; the reported MIP results already use bound propagation. We will explicitly mention this in the updated paper.
>
> 3. Thank you for the reference to this recent paper. We will consider these additional experiments.
>
> 4. The point you raise relates to BNNs in general, rather than to our particular work. BNNs are amenable to fast hardware implementations as in the papers [a-c], which are much harder to achieve for non-binarized networks. As such, we believe it is important to study the robustness of BNNs to attacks, regardless of whether there exists robust non-binarized counterparts of similar size.
>
> [a] Liang, Shuang, et al. "FP-BNN: Binarized neural network on FPGA." Neurocomputing 275 (2018): 1072-1086.
> [b] McDanel, Bradley, Surat Teerapittayanon, and H. T. Kung. "Embedded binarized neural networks." arXiv preprint arXiv:1709.02260 (2017).
> [c] Yang, Li, Zhezhi He, and Deliang Fan. "A Fully Onchip Binarized Convolutional Neural Network FPGA Impelmentation with Accurate Inference." Proceedings of the International Symposium on Low Power Electronics and Design. ACM, 2018.

---

> > ### Comment · AnonReviewer1 · 2018-12-09
> > **thanks for the revision**
> >
> > I thank the authors for the revision.
> >
> > Regarding 1, I think until the results on all the test images are published, I cannot recommend acceptance of the paper, because in my experience, the results can change significantly when testing on the entire test set versus a small subset of them.
> >
> > Regarding 4, I still find it difficult to understand the significance of BNNs when non binarized networks of much higher performance can be trained. I can certainly see ways to quantize non-binarized networks to facilitate hardware implementations.
> >
> > Given these shortcomings, I feel I am unable to change my rating for this paper.I would suggest that the authors revise and resubmit this paper with complete experimental results and a careful evaluation against non-binarized networks.

---

> > > ### Author Response · Authors · 2018-12-12
> > > **Thank you - A response**
> > >
> > > Dear reviewer, thanks for taking the time to read our revised paper.
> > >
> > > Regarding 1: The sample of 1,000 test points that we used shows clear trends. We report standard deviation/quantiles whenever possible to give a full view of the results. Given that we lay out the results clearly and discuss regimes where our methods perform well or not, we believe we should not be at a disadvantage due to limited computational resources. Additionally, given that we are presenting optimization methods, a sample of 1,000 instances is very consistent with the size of the benchmarks used in the optimization literature. For instance, in Mixed Integer Programming, the common benchmark of MIPLIB2010 has 361 instances (http://miplib2010.zib.de/) that are all *very different*, i.e. coming from various applications/generators. In our setting, our optimization problems are near identical: same mathematical formulation (variables, constraints) and very similar data (input images, epsilon). As such, results on 1,000 test images are amply representative of the behavior of the algorithms we analyze.
> > >
> > > Additionally, we would like to give a sense of the computational requirements for running experiments on the full 60,000 MNIST test points: for a single test point, method, network and epsilon, the method is run for 3 minutes. Considering the 6 values of epsilon, 8 network architectures and 4 methods (MIP, IPROP, PGD, SPSA), we would need 60K x 6 x 8 x 4 x 3 min > 34 million CPU minutes. Even with our cluster of 200 CPUs running simultaneously non-stop, we would need *120 days* to obtain the full results. We do agree that our initial 100 test points were too small a sample, and that is why we increased the sample size by one order of magnitude.
> > >
> > > Regarding 4: The practical relevance of binarized networks is studied at length in many papers. The recent set of contemporary papers on the topic by Courbariaux et al. and Hubara et al. (2016, 2017) are cited more than 1,000 times according to Google Scholar. The XNOR-Net paper on binarized convolutional networks is cited more than 800 times since ECML-PKDD 2016. As such, some researchers are now taking binarized networks to hardware implementations. Whether the binarized network was trained with -1/+1 weights from scratched or quantized from a full-precision network is irrelevant to our paper, as we take a trained network as input and attack it.

---

### Public Comment · (anonymous) · 2018-11-06
**Weak baselines**

While the motivation for studying attacks on binarized is not quite clear to me, I would like to point out that there are much stronger baselines than FGSM for attacking discrete, non-differentiable networks. In particular, several prior works have attempted to suggest binarization as a plausible defense and have evaluated their proposal by coming up with various attacks, all of which were subsequently broken because their attack method was weak compared to PGD [1] (and BPDA) [2]. So it is not sufficient to just compare against FGSM (as some reviewers have also pointed out).

[1] https://arxiv.org/abs/1706.06083
[2] https://arxiv.org/abs/1802.00420

---

> ### Author Response · Authors · 2018-11-14
> **Our FGSM is indeed PGD**
>
> Thank you for taking the time to read our paper!
>
> We just responded to the reviews with the following:
> "In our paper, FGSM refers to “iterated FGSM” or “multi-step FGSM” or PGD (these are all referring to the same method, e.g. see page 4 of https://arxiv.org/abs/1706.06083). We make that clear in section 2: “Soon thereafter, an iterative variant of FGSM was shown to produce much more effective attacks (Kurakin et al., 2016); it is this version of FGSM that we will compare against in this work.”. In fact, we run iterated FGSM/PGD for 3 minutes (same as MIP and IProp) with random restarts every 100 iterations. This provides FGSM with the same computational budget as IProp. We will update the paper to clarify this point in the experiments section."

---

> > ### Public Comment · ~Nicholas_Carlini1 · 2018-11-14
> > **Please use standard terminology**
> >
> > FGSM is a specific attack defined by Goodfellow et al. This attack takes one step in the direction of the gradient.
> >
> > If you are using a different attack---the "Basic Iterative Method" from Kurakin et al., say---then you should call it by that attack name, and don't call it FGSM. This is misleading.

---

> > > ### Author Response · Authors · 2018-11-14
> > > **Will do**
> > >
> > > Thanks for your comment.
> > >
> > > We agree: since the name Projected Gradient Descent (PGD) has been widely adopted to refer to the iterative version (as popularized in https://arxiv.org/abs/1706.06083, page 4), we will update the paper to use PGD throughout.

---

### Author Response · Authors · 2018-11-26
**Revised version of the paper**

We thank the reviewers for their comments and suggestions. We hope that the revised version of the paper and our direct replies to the reviews address all the issues that were raised.

In particular, we note the following changes in the revised version:

- PGD: We now refer to the competing gradient-based attack as Projected Gradient Descent (PGD), rather than FGSM, in all figures and the text. We would like to emphasize that all the results reported in the original submission are indeed for PGD, but we were using the name FGSM to refer to it. The reviewers have correctly suggested that PGD is the right name for the method we are using, given that it is iterative (as opposed to the one-step FGSM).

- Additional baselines: On Reviewer1's recommendation, we have compared against the "simultaneous perturbation stochastic approximation" (SPSA) method used in [*]. The comparison with IProp is in the Appendix. SPSA performs significantly worse than IProp on MNIST, as can be seen in Figure 7.

- Larger test subset: On Reviewer1's recommendation, we have run additional experiments that use 1,000 instead of 100 MNIST test images to strengthen the results. All MNIST figures in the revised version now use 1,000 test images. The results are qualitatively consistent with the original results we reported.

- Larger epsilon: On Reviewer1's recommendation, we have run both our IProp method and PGD with larger attack radii, namely epsilon={0.05, 0.1, 0.2}; Figure 2 shows the prediction flip rates for MNIST. For these large radii, fooling the neural network is relatively easy, as manifested by the high bars. PGD can outperform IProp in this easy regime since IProp is more computationally expensive.

- Notation for h variables: On Reviewer3’s suggestion, the h variables are now always in {-1,1}, including in the MIP formulation.

[*] Adversarial Risk and the Dangers of Evaluating Against Weak Attacks. https://arxiv.org/pdf/1802.05666.pdf.

---

### Meta-Review · Area_Chair1 · 2018-12-17
**Good paper, accept.**

**Confidence:** 4
**Recommendation:** Accept (Poster)

**Metareview:**

The paper provides a novel attack method and contributes to evaluating the robustness of neural networks with recently proposed defenses. The evaluation is convincing overall and the authors have answered most questions from the reviewers. We recommend acceptance.